# Physiological Mechanisms of Exogenous ABA in Alleviating Drought Stress in *Nitraria tangutorum*

**DOI:** 10.3390/plants14172643

**Published:** 2025-08-25

**Authors:** Xiaolan Li, Hanghang Liu, Cai He, Yi Li

**Affiliations:** College of Forestry, Gansu Agricultural University, Lanzhou 730070, China; lixiaolan0889@163.com (X.L.); liuhanghang131456@163.com (H.L.); hcyldfcl@163.com (C.H.)

**Keywords:** *Nitraria tangutorum*, drought stress, exogenous ABA, growth traits, physiology and biochemistry

## Abstract

Drought stress caused by continuous global warming poses a severe challenge to the growth and development of *Nitraria tangutorum*. Abscisic acid has an important regulatory function in the process of plants responding to drought stress. This study took the *N. tangutorum* seedlings of Zhangye provenance 2-17-16 genealogy as the research object to explore the physiological mechanism of how different concentrations of exogenous ABA alleviate drought damage in *N. tangutorum*. The results showed that exogenous ABA could promote the growth and increase the leaf relative water content of *N. tangutorum* seedlings under drought stress. It alleviates the photosynthetic inhibition phenomenon of *N. tangutorum* seedlings under drought stress by regulating the photoprotective mechanism and energy distribution efficiency of photosystem II. It also alleviates the drought damage of *N. tangutorum* by increasing the content of osmotic-adjustment substance contents such as soluble sugar, soluble protein, proline, and starch, as well as enhancing the activity of antioxidant enzymes such as POD, SOD, and CAT. The comprehensive analysis showed that 20 μM and 30 μM ABA have the best alleviating effects on the drought damage of *N. tangutorum* seedlings. This study provides a theoretical basis for the restoration, propagation, and protection of *N. tangutorum*, and it is of great significance for maintaining the balance and stability of desert ecosystems.

## 1. Introduction

*Nitraria tangutorum* is a perennial deciduous shrub of *Nitraria* in Nitrariaceae. It is mainly distributed in desert and semi-desert lake basin sandy lands, mountain front plain sand accumulation lands, and clay lands with wind-accumulated sands, and it is of great economic and ecological value [1]. As a typical constructive species in desert and semi-desert regions, *N. tangutorum* has the characteristics of drought resistance, high temperature resistance, cold resistance, salinity-alkali resistance, barrenness resistance, and wind and sand resistance [2]. Its plant height is usually 0.5–2 m, with branches lying prostrate or slanting upwards. The significantly shortened internode structure can minimize the overall transpiration surface area of the plant to the greatest extent. Its leaves are fleshy lanceolate in shape, covered with a waxy layer on the surface, which can effectively lock in water and reflect strong light. Its branches and leaves are rich in osmoregulatory substances such as proline and betaine, which can rapidly accumulate under water stress to maintain cell osmotic pressure balance and avoid damage to cells due to dehydration [3,4]. Its root system is well developed, with many branches and strong sprouting ability, and it can produce a large number of adventitious roots rapidly after being buried in sand and soil, thus forming new plants and expanding the thicket to form fixed and semi-fixed *N. tangutorum* coppice dunes, which can trap and stabilize large amounts of quicksand, preventing the expansion and erosion of deserts. *N. tangutorum*, as a pioneer tree species for windbreak and sand fixation in desert areas, plays an important role in maintaining the balance of regional ecosystems and improving the quality of the ecological environment [5]. The climate in desert areas is dry, precipitation is scarce and varies greatly in time and space, water resources are scarce, and the ecological environment is extremely fragile. Desert plants are highly sensitive to human activities and global climate change [6,7]. Over the past century, the drought stress caused by continuous global warming has posed a huge environmental challenge to desert flora [8,9].

Against the backdrop of global warming and dramatic shifts in precipitation patterns, extreme disasters occur frequently in many regions [10]. Among numerous extreme climate disaster events, drought is one of the most frequent and devastating natural disasters, especially in arid and semi-arid regions [11]. Drought stress not only affects the morphological characteristics of plants, but also induces a reduction in plant cell hydration and an imbalance in osmotic pressure, leading to stomatal closure and loss of photosynthetic capacity, which seriously affects the normal growth and development of plants [12]. Cell growth and division are the basis of plant growth and development, and the loss of water causes a decrease in cell turgor pressure, affecting the mitotic process and, thus, inhibiting plant growth. Wang et al. showed that drought stress significantly reduced growth indexes such as the plant height, aboveground dry weight, and leaf relative water content (RWC) of *Agropyron mongolicum* seedlings in an experiment to study the effects of drought stress on the growth characteristics of *Agropyron mongolicum* seedlings [13]. To cope with drought stress, plant cells accumulate osmoregulatory substances such as proline, soluble sugars, and starch to maintain cell turgor and water uptake capacity [14]. Xia et al. analyzed the combined effects of 24-Epibrassinolide and nitric oxide on potted kiwifruit seedlings under drought stress and indicated that drought stress would induce the accumulation of smoregulatory substances such as proline and soluble sugar [15]. In addition, plants also induce changes in a series of antioxidant enzyme activities, clearing excess reactive oxygen species (ROS) accumulated within cells, reducing the degree of membrane lipid peroxidation, and lowering malondialdehyde (MDA) content, thereby alleviating the oxidative damage caused by drought stress to plant cells [16]. Das et al. subjected drought-tolerant (TV-23) and -sensitive (S.3/A3) tea cultivars to drought stress for 21 days to explore the effects of drought stress on antioxidants in both tea cultivars. The results showed that the activities of catalase (CAT), peroxidase (POD), and superoxide dismutase (SOD) were enhanced in both tea varieties under drought stress, and it was considered that the enhancement level was directly proportional to drought tolerance [17]. In addition to accumulating osmoregulators, antioxidants, and reactive oxygen species (ROS) scavengers to cope with drought stress, plant hormones also play a pivotal role in regulating plant responses to various drought conditions.

Abscisic acid (ABA) is the most crucial phytohormone in plant responses to drought stress. Exogenous ABA treatment can induce stomatal closure, promote the accumulation of osmoregulatory substances, enhance the enzymatic antioxidant defense system, improve the photosynthetic performance and RWC of plants, promote plant growth, and ultimately increase the drought resistance of plants [18]. Wei et al. found that exogenous ABA significantly increased the tolerance of common wheat seedlings to drought stress by applying 10µM of ABA to wheat seedlings under PEG-stimulated drought stress [19]. Yang et al. used different concentrations of ABA to spray *Camellia oleifera* seedlings under drought stress and measured the related indexes. The results showed that appropriate concentrations of exogenous ABA could inhibit the photosynthetic rate, activate the activity of antioxidant enzymes, and promote the accumulation of osmoregulatory factors, thereby reducing oxidative damage, maintaining the stability of cell turgor pressure, and improving the drought resistance of *Camellia oleifera* seedlings [20]. Furthermore, Zhang et al. also showed in their research on the effects of ABA on the growth traits of two maize cultivars under drought stress that exogenous ABA could enhance the RWC and dry matter (DM) of both maize cultivars [21]. There are many research contents related to the effects of exogenous ABA on the physiological and biochemical characteristics of different species under drought stress, but there are few reports on *N. tangutorum*. As a pioneer tree species for windbreak and sand fixation in desert areas, water is the key factor restricting the normal growth and development of *N. tangutorum*. Continuous global warming has led to an intensification of drought in desert areas, making vegetation restoration increasingly difficult and accelerating the rate of desertification [22,23]. Therefore, we analyzed the physiological status of *N. tangutorum* seedlings under different degrees of drought stress and screened the drought background suitable for exogenous ABA treatment to avoid irreversible damage to *N. tangutorum* caused by extreme drought to *N. tangutorum*. We aimed to explore the effects of exogenous ABA on the growth traits, physiological characteristics, and chlorophyll fluorescence parameters of *N. tangutorum* under severe drought stress, and to clarify the mitigation mechanism of exogenous ABA on *N. tangutorum* under drought stress. This study provides a theoretical basis for the restoration, propagation, and protection of *N. tangutorum* in desert areas, and it is of great significance for the desertification control and maintenance of balance and stability of desert ecosystems.

## 2. Results

### 2.1. Effects of Different Degrees of Drought Stress on Growth Traits and Fluorescence Characteristic in N. tangutorum Seedlings

#### 2.1.1. Effects of Different Degrees of Drought Stress on the Plant Height and RWC of *N. tangutorum* Seedlings

As shown in Figure 1 (left), the height of *N. tangutorum* seedlings continued to grow under different levels of drought stress, but the growth rate showed a decreasing trend with the aggravation of drought stress. The plant height of *N. tangutorum* seedlings under D1, D2, D3, and D4 treatments decreased by 35.45%, 36.06%, 47.27%, and 59.39%, respectively, compared with D0. As shown in Figure 1 (right), the RWC of *N. tangutorum* seedlings decreased significantly compared with the control under different degrees of drought stress. Compared with D0, the RWC under D1, D2, D3, and D4 treatments decreased by 4.88%, 10.55%, 18.34%, and 25.84%, respectively.

#### 2.1.2. Effects of Different Degrees of Drought Stress on Fluorescence Characteristic in *N. tangutorum* Seedlings

Different degrees of drought stress led to the inactivation of the Photosystem II (PSII) reaction center, fluorescence leakage, obstruction of electron transfer, and reduction of light energy conversion efficiency. Minimum fluorescence (Fo) (Figure 2A) continuously increased with the intensification of drought stress levels. Compared with D0, the Fo increased by 18.39%, 25.55%, 35.68, and 54.85%, respectively, under D1, D2, D3, and D4 treatments. Maximum quantum yield of *PSⅡ* photochemistry (Fv/Fm) (Figure 2B) and effective quantum yield of *PSⅡ* (ΦPSⅡ) (Figure 2C) continuously decreased with the intensification of drought stress levels, and Fv/Fm and ΦPSII decreased by 30.4% and 44.06%, respectively, under D4 treatment compared with D0. Non-photochemical quenching (NPQ) is a photoprotective mechanism of plants, which dissipates heat energy to mitigate photooxidative damage. As shown in Figure 2D, NPQ shows an increasing and then a decreasing trend with the intensification of drought stress levels, and it is speculated that the decrease in NPQ values under D4 treatment may be due to the collapse of the photosystem as a result of the extreme drought that caused irreversible damage to the plants.

### 2.2. Effects of Different Degrees of Drought Stress on MDA Content in Different Tissues of N. tangutorum Seedlings

As shown in Figure 3A, the MDA content in the roots, stems, and leaves of *N. tangutorum* seedlings shows a continuous upward trend with the aggravation of drought. The MDA content in roots increased significantly under different degrees of drought stress compared with the control, and the MDA content in roots under D4 treatment was 3.23 times higher than that of D0. The content of MDA in stems and leaves increased significantly compared with D0 under D2, D3, and D4 treatments. The MDA content in stems and leaves under D4 treatment was 2.02 and 2.66 times that of D0, respectively.

### 2.3. Effects of Different Degrees of Drought Stress on Osmoprotectants Content in Different Tissues of N. tangutorum Seedlings

As shown in Figure 3B, the content of soluble sugar in the roots, stems, and leaves of *N. tangutorum* seedlings increased overall with intensification drought. The lower soluble sugar content in leaves under the D2 treatment than the D1 treatment may be due to experimental error. The soluble sugar content in roots and leaves increased significantly under different levels of drought stress compared with the control, and the soluble sugar content in roots and leaves was 6.26 and 4.30 times higher than that of D0 under the D4 treatment, respectively. The soluble sugar content in stems did not increase significantly under D1 treatment compared with D0 but increased significantly under other treatments compared with D0. The soluble sugar content in the stems under D4 treatment was 3.15 times higher than that of the D0.

As shown in Figure 3C, the soluble protein content in roots and stems of *N. tangutorum* seedlings was positively correlated with the degree of drought stress. The soluble protein content of roots increased significantly under D2, D3, and D4 treatments compared to D0, and the soluble protein content of stems increased significantly under all treatments compared to D0. Soluble protein content in roots and stems under D4 treatment increased by 45.09% and 50.93%, respectively, compared to D0. The soluble protein content in the leaves of *N. tangutorum* seedlings first increased and then decreased with the intensification of drought stress levels, reaching the maximum value under D3 treatment, which increased by 80.08% compared with the control.

As shown in Figure 3D, the proline content in the stems and leaves of *N. tangutorum* seedlings kept increasing with the intensification of drought stress levels. The proline content in the stems and leaves under D4 treatment was 1.7 and 1.9 times that of D0, respectively. The proline content in the roots of *N. tangutorum* seedlings increased first and then decreased with the intensification of stress degree, and it reached the maximum value under D3 treatment. The proline content in the D3 treatment was 4.29 times higher than that of D0.

### 2.4. Effects of Different Degrees of Drought Stress on Starch Content in Different Tissues of N. tangutorum Seedlings

As shown in Figure 4A, the starch content in the roots, stems, and leaves of *N. tangutorum* seedlings continuously decreased with the aggravation of drought stress. The starch content in roots of *N. tangutorum* seedlings decreased by 3.38%, 6.36%, 10.05%, and 29.30% under D1, D2, D3, and D4 treatments, respectively, compared with D0; the starch content in stems decreased by 24.72%, 33.36%, 27.69%, and 49.06%, respectively, compared with D0; and the starch content in leaves decreased by 8.90%, 40.17%, 68.70%, and 88.09%, respectively, compared with D0.

### 2.5. Effects of Different Degrees of Drought Stress on Antioxidant Enzymes Activity in Different Tissues of N. tangutorum Seedlings

As shown in Figure 4B, the POD activity in the roots, stems, and leaves of *N. tangutorum* seedlings continuously increased with the intensification of drought stress levels. Except for the D1 treatment in which POD activity in roots, stems, and leaves did not increase significantly compared with D0, POD activity in roots, stems, and leaves increased significantly compared with D0 under all treatments. The POD activity in the roots of *N. tangutorum* seedlings under the D2, D3, and D4 treatments was 2.15, 2.72, and 2.92 times higher than under the D0 treatment; the POD activity in stems was 1.83, 2.25, and 2.41 times higher than that of D0; the POD activity in leaves was 1.89, 2.64, and 2.8 times higher than that of D0.

It can be seen from Figure 4C that the SOD activity in the roots and leaves of *N. tangutorum* seedlings increased first and then decreased with the intensification of drought stress levels, reaching the maximum value under D3 treatment, respectively. Compared with D0, the SOD activities in the roots and leaves of *N. tangutorum* seedlings under the D3 treatment increased by 142.38% and 170.13%, respectively. The SOD activity in stems increased continuously with the increase in drought stress levels, and the SOD activity under the D4 treatment increased by 55.92% compared with D0.

It can be seen from Figure 4D that the CAT activity in the roots, stems, and leaves of *N. tangutorum* seedlings increased first and then decreased with the intensification of drought stress levels. In the early stage of drought stress, the CAT activity in roots, stems, and leaves increased continuously and began to decrease after reaching the maximum value under D2, D3, and D2 treatments, respectively. The CAT activity in the roots and leaves of *N. tangutorum* seedlings under the D2 treatment was 1.86 and 1.79 times higher, respectively, than under the D0 treatment, and the CAT activity in stems of *N. tangutorum* seedlings under the D3 treatment in stems of *N. tangutorum* seedlings was 1.48 times higher than under the D0 treatment.

### 2.6. Effects of Exogenous ABA on Growth Traits and Fluorescence Characteristics of N. tangutorum Seedlings Under Drought Stress

#### 2.6.1. Effects of Exogenous ABA on the Plant Height and RWC of *N. tangutorum* Seedlings Under Drought Stress

As shown in Figure 5 (left), different concentrations of exogenous ABA have different effects on the growth status of *N. tangutorum* seedlings under drought stress. Low concentrations of ABA promote the growth of *N. tangutorum* seedlings under drought stress, while high concentrations of ABA inhibit their growth. The DA2 treatment had the best effect on alleviating the growth of *N. tangutorum* seedlings under drought stress, and the plant height increment of *N. tangutorum* seedlings under DA2 treatment was 1.4 times that of the drought control. As shown in Figure 5 (right), spraying different concentrations of ABA on *N. tangutorum* seedlings under the same drought stress could increase the leaf relative water content. The leaf relative water content of *N. tangutorum* seedlings showed a trend of first increasing and then decreasing with the increase in ABA spraying concentration. The leaf relative water content of *N. tangutorum* seedlings reached the maximum value under DA2 treatment, which was 7.72% higher than that of the drought control.

#### 2.6.2. Effects of Exogenous ABA on Fluorescence Characteristic of *N. tangutorum* Seedlings Under Drought Stress

As can be seen from Figure 6, different concentrations of exogenous ABA treatment of *N. tangutorum* seedlings under drought stress could decrease Fo; increase Fv/Fm, ΦPSII, and NPQ to varying degrees; and alleviate the phenomenon of the photosynthesis inhibition of *N. tangutorum* seedlings under drought stress. As shown in Figure 6A, among the different concentrations of ABA treatments, Fo declined most significantly under DA2 treatment, which decreased by 14.76% compared to the drought control. As can be seen from Figure 6B,C, Fv/Fm and ΦPSII rebounded the most under DA2 treatment, with Fv/Fm and ΦPSII rising by 11.46% and 29.75%, respectively, under DA2 treatment compared with the drought control. As shown in Figure 6D, NPQ first increased and then decreased with the increase in ABA concentration, reaching the maximum value under DA3 treatment, which increased by 97.54% compared with the drought control.

### 2.7. Effects of Exogenous ABA on MDA Content in Different Tissues of N. tangutorum Seedlings Under Drought Stress

As shown in Figure 7A, the MDA content in roots, stems, and leaves of *N. tangutorum* seedlings under drought stress decreased continuously with the increase in exogenous ABA concentration. The MDA content in roots of *N. tangutorum* seedlings under DA1, DA2, DA3, and DA4 treatments decreased by 6.14%, 19.50%, 35.09%, and 42.17%, respectively; the MDA content in stems of *N. tangutorum* seedlings decreased by 7.39%, 15.85%, 27.52%, and 31.12%, respectively; and the MDA content in leaves of *N. tangutorum* seedlings decreased by 11.89%, 25.81%, 34.34%, and 40.96%%, respectively, compared with the drought control.

### 2.8. Effects of Exogenous ABA on Osmolytes Content in Different Tissues of N. tangutorum Seedlings Under Drought Stress

As shown in Figure 7B, different concentrations of exogenous ABA increased the soluble sugar content in roots, stems, and leaves of *N. tangutorum* seedlings under drought stress to different degrees. The soluble sugar content in the roots and leaves of *N. tangutorum* seedlings under drought stress showed a trend of increasing and then decreasing with the increase in ABA concentration and reached the maximum value under DA3 treatment. The soluble sugar content in roots and leaves under DA3 treatment was 1.29 and 1.60 times higher than that of drought control, respectively. The soluble sugar content in the stems of *N. tangutorum* seedlings rose with the increase in ABA concentration under drought stress, and the soluble sugar content in the stems under DA4 treatment was 2.04 times that of the drought control.

As shown in Figure 7C, different concentrations of exogenous ABA treatments could increase the soluble protein content in roots, stems, and leaves of *N. tangutorum* seedlings under drought stress. The content of soluble protein in different tissues under the same drought condition changed differently with the increase in ABA concentration. The soluble protein content in roots showed a trend of first increasing and then decreasing with the increase in ABA concentration, and it reached the maximum value under DA3 treatment, which increased by 41.23% compared with the drought control. The soluble protein content in stems and leaves increased continuously with the increase in ABA concentration, and the soluble protein content in stems and leaves under DA4 treatment increased by 32.75% and 39.79%, respectively, compared with the drought control.

As shown in Figure 7D, different concentrations of exogenous ABA treatments could increase the proline content in roots, stems, and leaves of *N. tangutorum* seedlings under drought stress to different degrees. The proline content in roots showed a trend of first increasing and then decreasing with the increase in ABA concentration, and it reached the maximum value under DA3 treatment, which increased by 21.90% compared with the drought control. The proline content in stems and leaves increased continuously with the increase in ABA concentration, and the proline content in stems and leaves under DA4 treatment increased by 25.13% and 50.82%, respectively, compared with the drought control.

### 2.9. Effects of Exogenous ABA on Starch Content in Different Tissues of N. tangutorum Seedlings Under Drought Stress

As shown in Figure 8A, different concentrations of exogenous ABA treatment have an effect on starch content in different tissues of of *N. tangutorum* seedlings under drought stress. Under drought stress, the starch content in the roots, stems, and leaves of *N. tangutorum* seedlings increased continuously with the increase in ABA concentration. The starch content of leaves significantly increased under the treatments of DA2, DA3, and DA4 compared with the drought control, while the starch content in other tissues showed no significant changes under various concentrations of ABA treatments. The starch content in the roots, stems, and leaves of *N. tangutorum* seedlings under DA4 treatment increased by 8.61%, 14.05%, and 18.79%, respectively, compared with the drought control.

### 2.10. Effects of Exogenous ABA on Antioxidant Enzymes Activity in Different Tissues of N. tangutorum Under Drought Stress

It can be seen from Figure 8B that different concentrations of exogenous ABA spraying treatments could increase the POD activity in roots, stems, and leaves of *N. tangutorum* seedlings under drought stress. The increase in POD activity in the roots, stems, and leaves of *N. tangutorum* seedlings is positively proportional to the concentration of ABA treatment. The POD activity in roots of *N. tangutorum* seedlings under DA1, DA2, DA3, and DA4 treatments increased by 17.16%, 38.42%, 54.51%, and 69.26%, respectively; the POD activity in stems of *N. tangutorum* seedlings increased by 16.22%, 31.36%, 50.51%, and 68.73%, respectively; and the POD activity in leaves of *N. tangutorum* seedlings increased by 10.87, 24.75, 39.69, and 62.09%, respectively, compared with the drought control.

It can be seen from Figure 8C that different concentrations of exogenous ABA treatments can increase the SOD activity in different tissues of *N. tangutorum* seedlings under drought stress to varying degrees. The SOD activity in roots of *N. tangutorum* seedlings under drought stress showed a trend of first increasing and then decreasing with the increase in ABA concentration and reached the maximum value under DA3 treatment, which was 1.28 times higher than that of the drought control. The SOD activities in stems and leaves rose with the increase in ABA concentration, and the SOD activities in both stems and leaves under DA4 treatment were 1.48 times higher than those of the drought control.

As shown in Figure 8D, the response of CAT activity in different tissues of *N. tangutorum* seedlings under drought stress to different concentrations of exogenous ABA treatment was different. The CAT activity in roots of *N. tangutorum* seedlings under drought stress increased and then decreased with the increase in exogenous ABA concentration and reached the maximum value under DA3 treatment, which was 24.48% higher than that of the drought control. The CAT activity in roots under DA4 treatment was lower than that of the drought control, and it was speculated that the high concentration of ABA treatment might inhibit the antioxidant defense system of *N. tangutorum* seedlings. The increase in CAT activity in stems and leaves of *N. tangutorum* seedlings under drought stress was directly proportional to the exogenous ABA concentration, and the CAT activity in stems and leaves under DA4 treatment increased by 8.09% and 22.59%, respectively, compared with the drought control.

## 3. Discussion

Drought stress disrupts the water balance of plant cells and inhibits their normal growth and development [24]. Plants regulate their own growth and alter the levels of osmoregulators and the activities of various antioxidant enzymes to cope with drought stress [25]. Drought stress leads to the outbreak of ROS, causing lipid peroxidation in biofilms, resulting in a large accumulation of harmful substances such as MDA [26]. This study, by exploring the effects of different degrees of drought stress on the MDA content in different tissues of *N. tangutorum* seedlings, indicated that the MDA content in the roots, stems, and leaves of *N. tangutorum* seedlings increased significantly with the aggravation of drought. RWC and plant height are the fastest morphological indicators for plants to respond to drought stress. The RWC and average height increment of *N. tangutorum* seedlings in this study decreased continuously with decreasing field capacity, which is consistent with Nouri et al. who concluded that drought stress severely inhibits the RWC and growth state of some pine species [27]. Chlorophyll fluorescence parameters are an ideal method to study and explore the effects of drought stress on plant photosynthesis [28]. In this study, drought stress destroyed the structure and function of the PSII reaction center of *N. tangutorum* seedlings, resulting in a decrease in ΦPSII and Fv/Fm and an increase in Fo, and subsequently triggering the photoprotection mechanism. The increase in NPQ consumed the excess heat energy to avoid oxidative damage. This is consistent with the changing trend of chlorophyll fluorescence parameters related to *L. ruthenicum* seedlings under different degrees of drought stress [29]. The downward trend of NPQ under extreme drought may be due to the irreversible damage to plants triggered by photosynthetic structures’ collapse caused by extreme drought. Osmoregulation is one of the important mechanisms for plants to cope with drought stress, and osmoregulatory substances can enable plant cells to maintain a certain turgor pressure under drought conditions to ensure the normal progress of plant physiological and metabolic activities [30]. This study determined the contents of osmoregulators in the roots, stems, and leaves of *N. tangutorum* seedlings under different degrees of drought stress, indicating that the contents of soluble sugars, soluble proteins, and proline in the roots, stems, and leaves of *N. tangutorum* seedlings continuously increased under mild to severe drought stress. It is similar to the results of a study by Gao et al. exploring the changes in the content of osmoregulatory substances in three provenances of *N. tangutorum* under different concentrations of PEG stress [31]. The decreasing trend in soluble protein in stems and proline content in roots under extreme drought in the present study is speculated to be due to proteolysis or oxidative damage. This change trend was also mentioned in the studies of Chen et al. on the soluble protein content of *P. orientalis* under drought stress [32] and Liang et al. on the proline content of mulberry under drought stress [33]. Starch is considered a determinant of plant adaptation to abiotic stresses [34]. There is evidence suggesting that there is a linear inverse relationship between drought stress severity and starch content [26]. Drought stress also activates the plant antioxidant defense system, effectively scavenging reactive oxygen species induced by adversity stress to reduce the damage caused by water deficit [35]. In this study, POD activity in the roots, stems, and leaves of *N. tangutorum* seedlings increased gradually with the intensification of drought stress levels, which is consistent with the findings of Zhou et al. who explored the physiological responses of *C. pauciflorus* to drought stress and concluded that the POD activity increased continuously with moderate to heavy drought stress [26]. Zhou et al. also showed that there were no significant differences in POD activity under mild drought stress compared to the control in their study on *C. pauciflorus*, and such results were also observed in our study. The activities of SOD and CAT first increased and then decreased with increasing drought severity, which aligns with the belief of Xiong et al. that the activities of SOD and CAT in oak under drought stress first increase and then decrease with the extension of stress time [36].

Plant hormones are important plant growth regulators, and exogenous application of plant hormones can enhance plant drought resistance by activating molecular and physiological defense systems [37,38]. Abscisic acid (ABA) is a sesquiterpenoid phytohormone widely regarded as the major regulatory factor of plant responses to water stress. Travaglia et al. demonstrated through their research on the application of exogenous ABA to wheat at different phenological periods under drought stress that exogenous ABA could mitigate the effects of drought stress on the growth and development of wheat plants [39]. In addition, Chen Juan, in her study on the effects of exogenous ABA on the physiology and growth of ginger plants under drought stress, showed that different concentrations of ABA had different effects on the growth and development of ginger under drought stress [40]. In this study, we explored the effects of different concentrations of exogenous ABA on plant height and leaf relative water content of *N. tangutorum* seedlings under drought stress, and we showed that different concentrations of exogenous ABA could increase the leaf relative water content of *N. tangutorum* seedlings, but the low concentration promoted plant growth and high concentrations inhibited plant growth. Our research results are consistent with those of Xi et al., who found that spraying low concentrations of ABA during the seedling stage can promote the growth of wheat seedlings, increase the relative water content of leaves, and thereby enhance their drought resistance when exploring the effects of exogenous ABA on drought resistance at different growth stages of wheat under drought stress [41]. This conclusion was also similarly validated in the study by Ma et al. on the response of *Onobrychis viciifolia* to exogenous ABA under drought stress [42]. The chlorophyll fluorescence parameters can indirectly reflect the strength of plant photosynthetic ability and non-destructively assess the state of PSⅡ [43]. In this study, the *N. tangutorum* seedlings under drought stress were treated with different concentrations of exogenous ABA, and related chlorophyll fluorescence parameters were determined. The results showed that the Fo decreased and the Fv/Fm, NPQ, and PSII increased to varying degrees compared to the drought control in *N. tangutorum* seedlings under drought stress at different concentrations of exogenous ABA treatment. It is indicated that exogenous ABA can make the energy absorbed by PSII antenna pigments flow more to the photochemical part, avoiding the dissipation in the form of fluorescence, so that *N. tangutorum* seedlings can maintain a higher photochemical efficiency under drought conditions, which is conducive to the progress of photosynthesis [44]. In exploring the effects of exogenous ABA on the content of osmoregulatory substances in 2-year-old *C. Reticulata* Zipao seedlings under drought stress, Wang et al. demonstrated that exogenous ABA could elevate the content of soluble sugars, soluble proteins, and proline in *C. Reticulata* Zipao seedlings under drought stress [45]. The results of this study on the effects of different concentrations of exogenous ABA on the content of osmoregulatory substances in *N. tangutorum* seedlings under drought stress were consistent with those of Wang et al., and exogenous ABA could increase the contents of soluble sugar, soluble protein, and proline in different tissues of *N. tangutorum* seedlings under drought stress. Few studies have involved the effect of exogenous ABA on the starch content of plants under drought stress in research on the physiological responses of exogenous ABA to plants under drought stress. In this study, the starch content in the roots, stems, and leaves of *N. tangutorum* seedlings under drought stress decreased under different concentrations of exogenous ABA, which may be due to the decomposition of starch into soluble sugar to enhance the osmoregulation ability of *N. tangutorum* seedlings under drought stress [46]. Lu et al.’s study on triploid bermudagrass under drought stress showed that the exogenous application of ABA could alleviate the oxidative damage caused by drought stress to triploid bermudagrass by increasing the leaf relative water content, increasing antioxidant enzymes activity, and decreasing malondialdehyde content [47]. Wang et al. also believed that exogenous ABA could reduce membrane permeability, improve antioxidant enzyme activity, and alleviate drought damage in different tissues of plants in their study of the physiological response of Kiwifruit potted seedlings to 60 µM exogenous ABA under drought stress [48]. This study explored the effects of different concentrations of exogenous ABA on the contents of malondialdehyde and activities of antioxidant enzymes in *N. tangutorum* seedlings under drought stress. The results showed that different concentrations of exogenous ABA treatments reduce the MDA content and increase the antioxidant enzyme activities in the roots, stems, and leaves of *N. tangutorum* seedlings. However, the CAT activity in the roots of *N. tangutorum* seedlings under drought stress was significantly lower than the drought control under 50 µM/L exogenous ABA treatment, which may be related to the conclusion that the high concentration of ABA inhibits the antioxidant defense system and aggravates the oxidative damage, as reported by Li Shanshan [49]. In addition, Li Shanshan also demonstrated through testing that low-concentration ABA pretreatment effectively alleviated the inhibitory effect of drought stress on the germination of *N. tangutorum* seeds and significantly increased their germination rate under drought stress [49].

## 4. Materials and Methods

### 4.1. Plant Materials

The experimental materials were the 2-17-16 genealogy of Zhangye provenance of *N. tangutorum*, provided by the Gansu Province Academy of Qilian Water Resource Conservation Forests Research Institute. In April 2024, 200 full and uniform *N. tangutorum* seeds were screened, soaked in concentrated sulfuric acid for 30 min, treated with 75% alcohol for 5 min, treated with 20% sodium hypochlorite for 12 min, and rinsed with distilled water 3–4 times. They were planted in special breeding trays for forest trees filled with nutrient soil and placed in plastic greenhouses of Gansu Agricultural University for seedling rearing. When the *N. tangutorum* seedlings’ height reached 5 ± 1 cm, they were transferred into plastic flowerpots with a diameter of 13 cm and a height of 14.5 cm for cultivation. The culture medium was composed of completely air-dried sand and soil mixed in a 1:1 (*v*/*v*) ratio, with a maximum water holding capacity of 26.84%. Each pot was filled with 700 g of soil, and one seedlings was transplanted. Standard pot-culture management was carried out for the seedlings to ensure they grow vigorously.

### 4.2. Experimental Design

#### 4.2.1. Drought Stress Treatments

In July 2024, *N. tangutorum* seedlings with uniform growth and a plant height of 20 ± 1 cm were selected for different levels of drought stress. The growth characteristics and physiological and biochemical changes of *N. tangutorum* seedlings under different degrees of drought stress were analyzed to determine the drought background suitable for exogenous ABA treatment. The experiment was conducted in five groups with one control (DO) and four drought gradients (D1, D2, D3, and D4) using 55–60%, 30–35%, 25–30%, 20–25%, and 15–20% of field water-holding capacities as treatment intervals, respectively. There were eight seedlings per group, totaling 40 seedlings. After determining that the field water-holding capacity of each group seedlings reached the corresponding stress range by the weighing method, the plant height was measured, and drought stress was carried out. At 18:00 every day, the moisture content of the test materials was replenished to the corresponding drought stress range by the weighing method. After continuous treatment for 10 days, the plant height, chlorophyll fluorescence parameters, and leaf relative water content were measured, and the roots, stems, and leaves of the *N. tangutorum* seedlings were pooled and sampled, respectively. After being quick-frozen in liquid nitrogen, the samples were stored in a −80 °C refrigerator for subsequent determination of physiological and biochemical indicators, with three biological replicates for each physiological and biochemical parameters.

#### 4.2.2. Abscisic Acid (ABA) Spray Treatments

The field holding capacities of 55–60% (D0) and 20–25% (D3) were used as the control and drought background for spray treatments with different concentrations of ABA, respectively, to explore the alleviation mechanism of drought damage caused by different concentrations of exogenous ABA in *N. tangutorum*. *N. tangutorum* seedlings of 20 ± 1 cm were selected and used to set up 7 treatment groups as normal watering control (CK), D3 drought-stressed control (CKD), D3 + water spray control (DW), D3 + 10 μM ABA (DA1), D3 + 20 μM ABA (DA2), D3 + 30 μM ABA (DA3), and D3 + 50 μM ABA (DA4). There were eight seedlings per group, totaling 56 seedlings. After the test materials reached the corresponding field water-holding capacity, the plant height of *N. tangutorum* was measured, and different concentrations of ABA spray treatments were carried out. The standard of ABA spray is that water droplets cover the front and back sides of the leaves without falling off. Spraying was carried out at 9:00 in the morning and once every two days. At 18:00 every day, the moisture content of the test materials was replenished to the corresponding drought stress range by the weighing method. After continuous treatment for 10 days, the plant height, chlorophyll fluorescence parameters, and leaf relative water content were measured, and the roots, stems, and leaves of the *N. tangutorum* seedlings were pooled and sampled, respectively. After being quick-frozen in liquid nitrogen, the samples were stored in a −80 °C refrigerator for subsequent determination of physiological and biochemical indicators, with three biological replicates for each of the physiological and biochemical parameters.

### 4.3. Measurement of Growth Index

A straightedge was used to measure the vertical distance from base to the apex of each seedling before and after treatment to calculate plant height increment. Three seedlings were randomly selected in each treatment, and all the leaves at two-thirds of the plant height were removed and weighed immediately (FW). Subsequently, the leaves were soaked in distilled water for 24 h and then fished out, and the turgid weight (TW) was determined after absorbing the water with filter paper. Finally, the dry weight (DW) of the leaves was measured after drying in an oven at 80 °C for 48 h. The leaf relative water content (RWC) of all treatments was calculated according to the following formula and expressed as a percentage [50].$$\mathrm{RWC}(\%)=\frac{FW-DW}{TW-DW}\times 100$$

### 4.4. Measurement of Chlorophyll Fluorescence Parameters

After each treatment was completed, chlorophyll fluorescence parameters such as minimum fluorescence (Fo), maximum quantum yield of *PSⅡ* photochemistry (Fv/Fm), effective quantum yield of *PSⅡ* (ΦPSⅡ), and non-photochemical quenching (NPQ) of the fourth fully expanded leaf of *N. tangutorum* seedlings in each treatment group were measured by using a FluorPen FP110 hand-held chlorophyll fluorometer (Czech Republic, Photon Systems Instruments) from 9:00 am to 11:00 am. Among them, the Fo and Fv/Fm values were measured after 30 min of dark treatment of the leaves [51].

### 4.5. Determination of Malondialdehyde (MDA) Content

The content of malondialdehyde was determined with reference to the experimental method of Huang et al. [52]. The 0.2 g frozen sample was homogenized with 3 mL of 5% trichloroacetic acid, and the crude extract was centrifuged at 12,000× *g* for 20 min to obtain the MDA supernatant. Then, 2 mL of supernatant and 2 mL of 0.6% thiobarbituric acid were mixed thoroughly and incubated in a water bath at 100 °C for 30 min. After rapid cooling, the mixture was centrifuged at 12,000 rpm for 10 min at 4 °C. Absorbance of the solution at 400, 500, and 600 nm was measured, and the MDA content was calculated and expressed as nmol·g^−1^ FW.

### 4.6. Determination of Osmotic Adjustment Substances Content

#### 4.6.1. Proline

The proline content of each treated sample was determined with reference to the experimental method of Hou et al. [53]. The 0.2 g of frozen sample was taken and homogenized with 5 mL of 3% (*w*/*v*) sulfosalicylic acid solution. The crude extract was heated in a boiling water bath for 10 min, and the supernatant was separated after cooling and centrifugation at 12,000 rpm for 15 min to obtain the proline extract. We thoroughly mixed 2 mL of proline extract, 2 mL of glacial acetic acid, and 2 mL of 2.5% acidic ninhydrin solution, and then we placed them in a boiling-water bath and heated for 30 min. After cooling to room temperature, we added 4 mL of toluene and shook thoroughly for extraction. The optical density at a wavelength of 520 nm was measured using a UV spectrophotometer, and the proline concentration was calculated based on the standard curve.

#### 4.6.2. Soluble Sugar

The content of soluble sugar in the samples was determined with reference to the experimental method of Wang et al. [54]. We took 0.2 g of frozen sample and added a little quartz sand and 10 mL of distilled water for grinding. The homogenate was extracted in a boiling water bath for 20 min and then centrifuged at 8000 r/min for 10 min after cooling. We took 0.5 mL of the supernatant, added 1 mL of anthracone reagent, and kept it warm in a 95 °C water bath for 10 min. After cooling to room temperature, the absorbance was recorded at a wavelength of 620 nm. Finally, we referred to the standard curve and calculated the content of soluble sugar.

#### 4.6.3. Starch

The residue after extraction of soluble sugars was dried in an oven at 80 °C and used for starch extraction. We added 10 mL of distilled water to the residue and then boiled for 15 min in a boiling-water bath. After cooling, 2 mL of 9.2 mol·L^−1^ perchloric acid was added and extracted in a boiling water bath for 15 min, cooled, and centrifuged. The extraction was repeated three times, and the supernatant was collected into a 50 mL volumetric flask; we made up the volume with distilled water. After cooling, the absorbance was recorded at a wavelength of 620 nm, and starch content was calculated according to the soluble sugar standard curve multiplied by 0.9 [55].

#### 4.6.4. Soluble Protein

The content of soluble protein in the sample tissue was determined with reference to the experimental method of Wang et al. [56]. We took 0.2 g of the frozen sample and put it in a mortar, and we added 8 mL of distilled water and a small amount of quartz sand for grinding. The crude extract was transferred to a clean centrifuge tube and centrifuged at 12,000 rpm for 10 min. We mixed 0.5 mL of supernatant, 0.5 mL of distilled water, and 5 mL of Coomassie Brilliant Blue G-250 reagent in the same tube to determine the absorbance at 595 nm. We calculated the content of soluble protein based on the standard curve.

### 4.7. Determination of Antioxidant Enzyme Activity

We took 0.2 g of the sample, placed it in a pre-chilled mortar, and grinded it with 5 mL of PBS (50 mM, pH 7.8). The homogenate was centrifuged at 12,000 r/min at 4 °C for 20 min, and the supernatant was used for the detection of different enzyme activities. Superoxide dismutase activity (SOD) was determined at 450 nm using the nitroblue tetrazolium method with reference to Sanyal et al. [57]. The activity of superoxide dismutase (POD) was determined at 470 nm by the guaiacol method [58]. Catalase activity (CAT) was assessed by monitoring the rate of H_2_O_2_ decomposition according to the research program of Gou et al. [59]. All enzyme activities were recorded in units of U·g^−1^·min^−1^ FW [60,61,62].

## 5. Conclusions

(1)Different levels of drought stress inhibited the growth, reduced the leaf relative water content, destroyed the reaction center structure of PSII, damaged the biofilm system, and led to a significant increase in the MDA content of *N. tangutorum* seedlings. *N. tangutorum* seedlings responded to drought stress by increasing the contents of osmoregulators such as soluble sugar, soluble protein, proline, and starch, as well as enhancing the activities of antioxidant enzymes such as POD, SOD, and CAT. The NPQ value, proline content in roots, and CAT activity in roots, stems, and leaves of *N. tangutorum* seedlings under D4 treatment were significantly lower than those under D3 treatment. It is speculated that extreme drought may cause metabolic collapse, resulting in irreversible damage to the *N. tangutorum* seedlings. After comprehensive analysis, D3 treatment (20–25% field capacities) was selected as the drought background of exogenous ABA treatment.(2)Different concentrations of exogenous ABA treatment of *N. tangutorum* seedlings under drought stress could promote the growth, increase the leaf relative water content, and alleviate the photosynthetic inhibition phenomenon of *N. tangutorum* seedlings under drought stress. In addition, exogenous ABA can also alleviate the drought damage of *N. tangutorum* seedlings under drought stress by increasing the contents of osmoregulators such as soluble sugar, soluble protein, proline, and starch, as well as enhancing the activities of antioxidant enzymes such as POD, SOD, and CAT. The comprehensive analysis showed that the exogenous ABA concentrations of 20 μM and 30 μM had the best alleviating effect on the drought damage of *N. tangutorum* seedlings. The determination of this concentration range is of significant guidance for the restoration, propagation, and conservation of *N. tangutorum* under in situ desert conditions. For example, applying this concentration of ABA during the seedling establishment stage can help seedlings better adapt to arid environments and improve establishment survival rates. Additionally, applying ABA to established *N. tangutorum* populations prior to extreme drought events can enhance their drought resistance, reducing growth inhibition and yield decline caused by drought.

## Figures and Tables

**Figure 1 plants-14-02643-f001:**
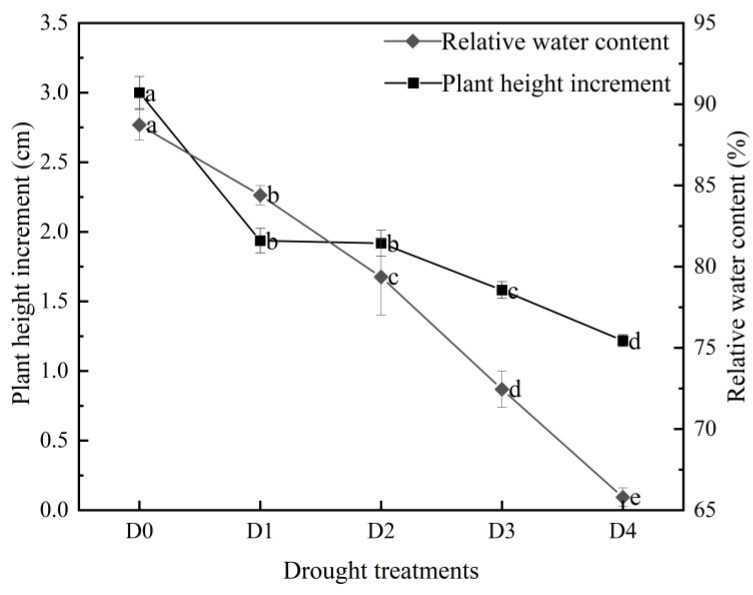
Effects of different degrees of drought stress on the plant height increment (left) and leaf relative water content (right) of *N. tangutorum* seedlings. D0 represents normal watering; D1, D2, D3, and D4 represent drought treatments with a field water-holding capacity of 30–35%, 25–30%, 20–25%, and 15–20%, respectively. Different lowercase letters beside the error bars indicate statistically significant differences between different treatments (ANOVA, *p* < 0.05), and the error bar shows the standard error (SE).

**Figure 2 plants-14-02643-f002:**
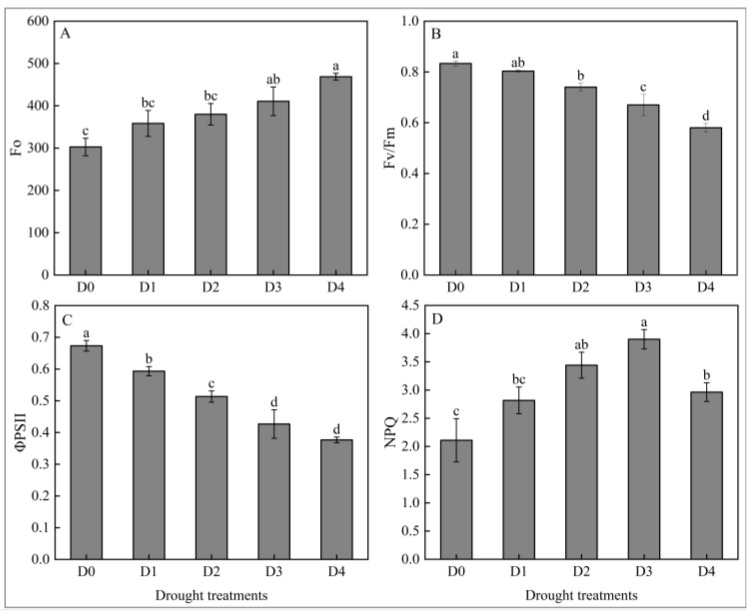
Effects of different degrees of drought stress on the chlorophyll fluorescence characteristics of *N. tangutorum* seedlings. (**A**) Fo; (**B**) Fv/Fm; (**C**) ΦPSII; (**D**) NPQ. D0 represents normal watering; D1, D2, D3, and D4 represent drought treatments with a field water-holding capacity of 30–35%, 25–30%, 20–25%, and 15–20%, respectively. Different lowercase letters above the error bars indicate statistically significant differences between different treatments (ANOVA, *p* < 0.05), and the error bar shows standard error (SE).

**Figure 3 plants-14-02643-f003:**
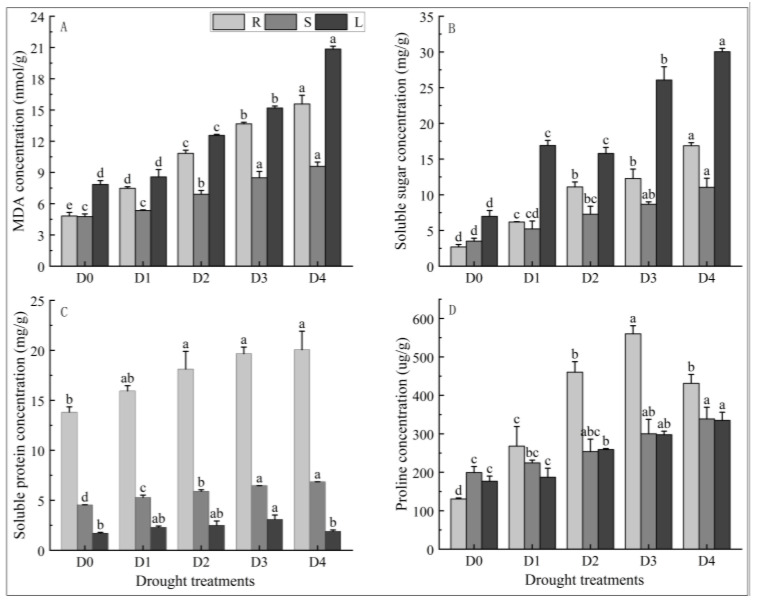
Effects of different degrees of drought stress on the contents of MDA and osmolytes in different tissues of *N. tangutorum* seedlings. (**A**) MDA content; (**B**) soluble sugar content; (**C**) soluble protein content; (**D**) proline content. D0 represents normal watering; D1, D2, D3, and D4 represent drought treatments with a field water-holding capacity of 30–35%, 25–30%, 20–25%, and 15–20%, respectively. R, S, and L represent roots, stems, and leaves, respectively. Different lowercase letters above the error bars indicate statistically significant differences among different treatments of the same tissue (ANOVA, *p* < 0.05), and the error bar shows standard error (SE).

**Figure 4 plants-14-02643-f004:**
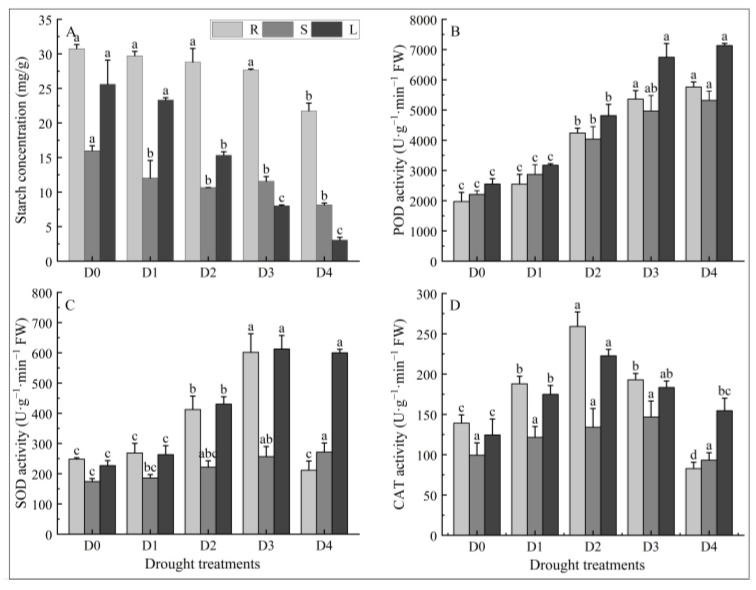
Effects of different degrees of drought stress on the starch content and antioxidant enzyme activity in different tissues of *N. tangutorum* seedlings. (**A**) Starch content; (**B**) POD activity; (**C**) SOD activity; (**D**) CAT activity. D0 represents normal watering; D1, D2, D3, and D4 represent drought treatments with a field water-holding capacity of 30–35%, 25–30%, 20–25%, and 15–20%, respectively. R, S, and L represent roots, stems, and leaves, respectively. Different lowercase letters above the error bars indicate statistically significant differences among different treatments of the same tissue (ANOVA, *p* < 0.05), and the error bar shows the standard error (SE).

**Figure 5 plants-14-02643-f005:**
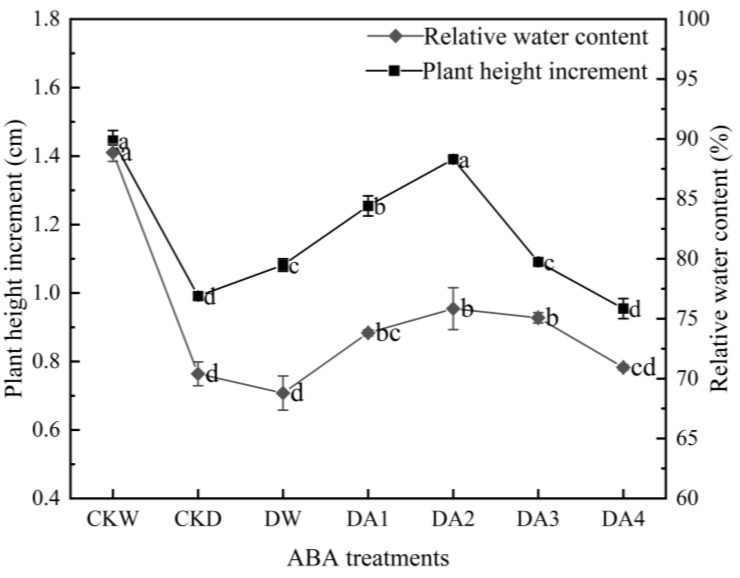
Effects of different concentrations of exogenous ABA on the height increment (left) and relative water content of leaves (right) of *N. tangutorum* seedlings under drought stress. CKW, CKD, and DW represent the normal watering control, D3 drought-stressed control, and D3 + water spray control, respectively; DA1, DA2, DA3, and DA4 represent the D3 + 10 μM, D3 + 20 μM, D3 + 30 μM, and D3 + 50 μM ABA spray treatments, respectively. Different lowercase letters beside the error bars indicate statistically significant differences between different treatments (ANOVA, *p* < 0.05), and the error bar shows the standard error (SE).

**Figure 6 plants-14-02643-f006:**
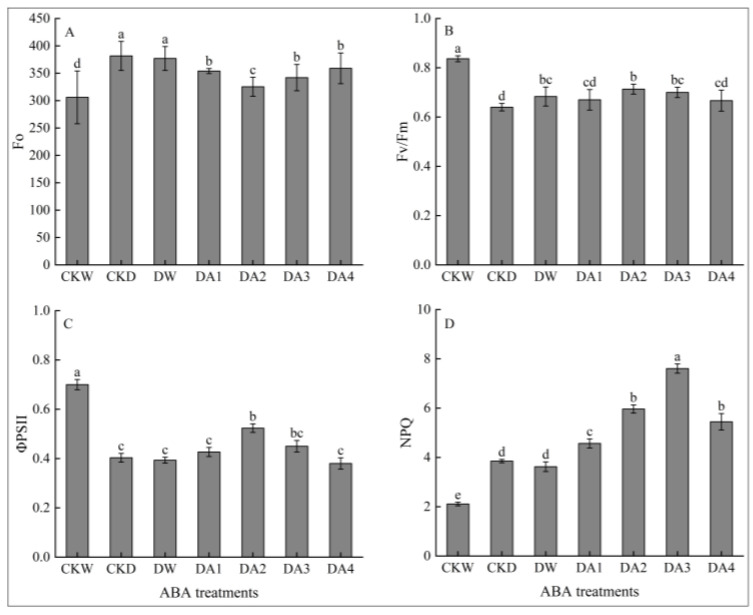
Effects of different concentrations of exogenous ABA on the chlorophyll fluorescence characteristics of *N. tangutorum* seedlings. (**A**) Fo; (**B**) Fv/Fm; (**C**) ΦPSII; (**D**) NPQ. CKW, CKD, and DW represent the normal watering control, D3 drought-stressed control, and D3 + water spray control, respectively; DA1, DA2, DA3, and DA4 represent the D3 + 10 μM, D3 + 20 μM, D3 + 30 μM, and D3 + 50 μM ABA spray treatments, respectively. Different lowercase letters above the error bars indicate statistically significant differences between different treatments (ANOVA, *p* < 0.05), and the error bar shows the standard error (SE).

**Figure 7 plants-14-02643-f007:**
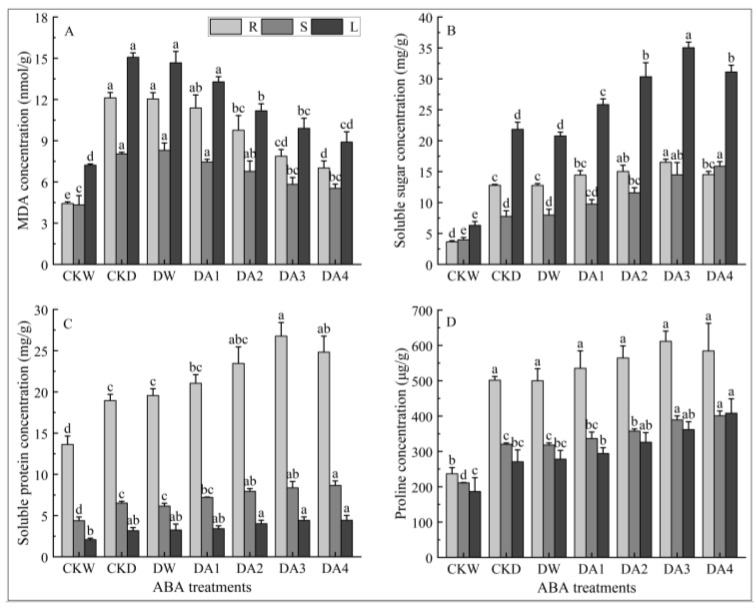
Effects of different concentrations of exogenous ABA on the contents of MDA and osmolyte in different tissues of *N. tangutorum* seedlings under drought stress. (**A**) MDA content; (**B**) soluble sugar content; (**C**) soluble protein content; (**D**) proline content. CKW, CKD, and DW represent the normal watering control, D3 drought-stressed control, and D3 + water spray control, respectively; DA1, DA2, DA3, and DA4 represent the D3 + 10 μM, D3 + 20 μM, D3 + 30 μM, and D3 + 50 μM ABA spray treatments, respectively. R, S, and L represent roots, stems, and leaves, respectively. Different lowercase letters above the error bars indicate statistically significant differences among different treatments of the same tissue (ANOVA, *p* < 0.05), and the error bar shows the standard error (SE).

**Figure 8 plants-14-02643-f008:**
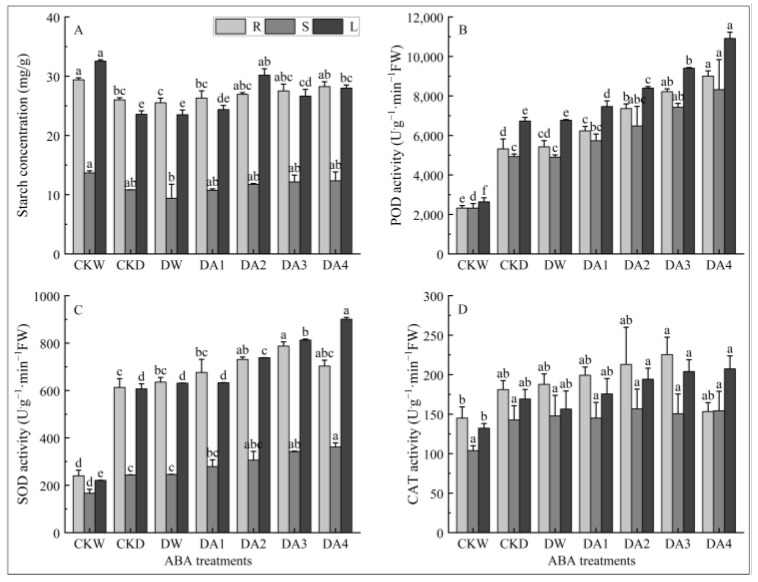
Effects of different concentrations of exogenous ABA on the content of starch and activity of antioxidant enzyme in different tissues of *N. tangutorum* seedlings under drought stress. (**A**) Starch content; (**B**) POD activity; (**C**) SOD activity; (**D**) CAT activity. CKW, CKD, and DW represent the normal watering control, D3 drought-stressed control, and D3 + water spray control, respectively; DA1, DA2, DA3, and DA4 represent the D3 + 10 μM, D3 + 20 μM, D3 + 30 μM, and D3 + 50 μM ABA spray treatments, respectively. R, S, and L represent roots, stems, and leaves, respectively. Different lowercase letters above the error bars indicate statistically significant differences among different treatments of the same tissue (ANOVA, *p* < 0.05), and the error bar shows the standard error (SE).

## Data Availability

All relevant data are within the article.

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
