# Peer review of "Physiological Mechanisms of Exogenous ABA in Alleviating Drought Stress in Nitraria tangutorum"

_plants, 2025, doi:10.3390/plants14172643_

Round 1
Reviewer 1 Report
Comments and Suggestions for Authors
This paper provides valuable insights into ABA's potential applications in plant stress physiology. While the findings contribute meaningfully to the field, several methodological and presentational aspects require attention to strengthen the manuscript's conclusions. Below are my major concerns and recommendations.
1. This species belongs to a salt- and alkali-tolerant, drought-resistant plant found at the edges of deserts. Is it appropriate to select it as the research material for studying ABA's drought mitigation mechanisms? Why not use a drought-sensitive material instead?
2.The Introduction section places excessive emphasis on the applied value of the plant, while providing insufficient discussion on its botanical characteristics relevant to the study in this manuscript.
3.If results are presented first, all abbreviations must be defined in full upon their first occurrence in the text.
4.The chlorophyll fluorescence measurements included too few parameters. Please provide all key indicators relevant to the analysis.
5.The Results section should present data objectively without interpretation. Please remove the speculative statements in lines 138-139.
6.The different treatments represent stress intensity levels rather than temporal sequences. Therefore, trend-based descriptions like 'increased initially then decreased' (which imply time-dependent changes) should be avoided when characterizing treatment effects. Please use more precise terminology that reflects stress-response relationships.
7.There appear to be multiple inconsistencies in the significance labeling throughout the results. A careful re-examination of all statistical annotations against the original analyses is strongly recommended.
8.Given the study's focus on mitigation effects, the current separation of stress and mitigation responses into distinct sections creates artificial fragmentation. A unified presentation comparing stress-with-mitigation versus stress-alone conditions would better address the core hypothesis.
9.The sample size (6-8 plants per treatment) appears insufficient for robust statistical analysis. Additionally, the experimental replication scheme (biological/technical replicates) needs clarification.
10.The soil moisture level of 55-60% appears too high to be considered a true drought stress treatment in this experimental system. Please justify this threshold or consider adjusting to a lower moisture level that would more realistically induce drought stress.
11.The drought treatment is inconsistently labeled as D1 (55-60% moisture) in the stress section but referred to as D0 in the ABA mitigation experiment. Please standardize the nomenclature throughout to prevent reader confusion. All sections should use either D0 or D1 consistently for this moisture level.
12.The experimental design limits causal interpretation of ABA effects, as ABA was only applied to drought-stressed plants. Without parallel ABA treatment in control (well-watered) conditions, observed changes cannot be conclusively attributed to drought-specific ABA responses rather than general ABA effects.
Author Response
|
Response to Reviewer 1 Comments |
|
Dear Reviewer: Thank you very much for taking the time to review this manuscript entitled “Physiological mechanisms of exogenous ABA in alleviating drought stress in Nitraria tangutorum” (ID: plants-3800017). Those comments are all valuable and very helpful for revising and improving our paper, as well as the important guiding significance to our researches. We have carefully revised the manuscript and provided the point-by-point response below. The changes in the revised manuscript have been highlighted in red. We hope these changes will strengthen our manuscript. |
|
Comments 1: This species belongs to a salt- and alkali-tolerant, drought-resistant plant found at the edges of deserts. Is it appropriate to select it as the research material for studying ABA's drought mitigation mechanisms? Why not use a drought-sensitive material instead? |
|
Response 1: We appreciate your insightful question regarding the selection of research materials. Drought-sensitive materials are prone to cellular damage under drought stress, leading to metabolic disorders, which disrupt the association between ABA and osmotic substances and mask the normal regulatory process of ABA. In contrast, N. tangutorum has developed an efficient and specific ABA regulatory network during its long-term evolution. Under drought stress, it can rapidly induce ABA synthesis and activate downstream protective mechanisms, which is more conducive to analyzing the drought alleviation mechanism of ABA. Additionally, as one of the primary tree species for ecological restoration, the research findings on N. tangutorum can directly guide ecological restoration in arid regions. |
|
Comments 2: The Introduction section places excessive emphasis on the applied value of the plant, while providing insufficient discussion on its botanical characteristics relevant to the study in this manuscript. |
|
Response 2: We sincerely appreciate the reviewer’s insightful observation regarding the layout of the Introduction. We have deleted the statement about applied value and added discussion related to botanical characteristics in the Introduction (see revised lines 24-31). |
|
Comments 3: If results are presented first, all abbreviations must be defined in full upon their first occurrence in the text. |
|
Response 3: We appreciate your careful reading and valuable comments. We have reviewed all abbreviations that first appear in the results and supplemented their full names upon their initial occurrence. Make sure that each abbreviation is clearly defined when it is first mentioned (see revised lines 102-108). |
|
Comments 4: The chlorophyll fluorescence measurements included too few parameters. Please provide all key indicators relevant to the analysis. |
|
Response 4: We sincerely appreciate your valuable comments. In our experimental design, we focused on the core functions and protective mechanisms of photosystem II. After reviewing relevant literature, we found that parameters such as Fo, Fv/Fm, ΦPSII and NPQ constitute a complete photosynthetic function evaluation system, which can support our research objectives. Therefore, we only measured these four parameters. We understand your concern and apologize for the lack of additional parameters. Due to conditionality constraints, we are unable to supplement extra chlorophyll fluorescence parameters at this stage. However, we confirm that the existing parameters are sufficient to support the key conclusions regarding the changes in photoprotection of N. tangutorum. We hope this statement can be approved by you. |
|
Comments 5: The Results section should present data objectively without interpretation. Please remove the speculative statements in lines 138-139. |
|
Response 5: We sincerely appreciate your suggestion. We have removed the relevant speculative statements as suggested. The Results section now solely presents objective data without any interpretive or speculative content (see revised lines 138-139). |
|
Comments 6: The different treatments represent stress intensity levels rather than temporal sequences. Therefore, trend-based descriptions like 'increased initially then decreased' (which imply time-dependent changes) should be avoided when characterizing treatment effects. Please use more precise terminology that reflects stress-response relationships. |
|
Response 6: We appreciate your valuable suggestion regarding the terminology for describing results. Following your guidance, we have carefully revised all relevant descriptions in the manuscript to avoid implying temporal sequences (see revised lines 104, 107, 110, 138, 141, 159, 166, 169, 172, 326). |
|
Comments 7: There appear to be multiple inconsistencies in the significance labeling throughout the results. A careful re-examination of all statistical annotations against the original analyses is strongly recommended. |
|
Response 7: We sincerely appreciate your careful review, which has helped us identify issues in the significance labels. Upon a thorough re-examination, we confirm that there were inappropriate significance labels in Figures 6A and 6B. We have corrected these to align with the original statistical analyses (see revised Figures 6A and 6B). |
|
Comments 8: Given the study's focus on mitigation effects, the current separation of stress and mitigation responses into distinct sections creates artificial fragmentation. A unified presentation comparing stress-with-mitigation versus stress-alone conditions would better address the core hypothesis. |
|
Response 8: We sincerely appreciate your professional review of our manuscript. We actually designed the group of D3 drought-stressed control (CKD) and D3 +different concentrations ABA spray treatments (DA1 / DA2 / DA3 / DA4) in the section of Abscisic Acid (ABA) spray treatments of Experimental design. This allowed us to comparatively analyze the mitigation effects of the ABA spray treatments (stress-with-mitigation) to the drought-stressed (stress-alone) (see revised lines 416-418). |
|
Comments 9: The sample size (6-8 plants per treatment) appears insufficient for robust statistical analysis. Additionally, the experimental replication scheme (biological/technical replicates) needs clarification. |
|
Response 9: Thank you for your valuable comment. Prior to the formal experiment, we conducted a systematic observation on the physiological index stability and individual differences of a large number of samples (far exceeding 8 plants) through preliminary pre-experiments. We found that this species exhibited high consistency in physiological responses under identical treatment conditions, and the individual fluctuation range was at an acceptable level. Based on these results and the actual sample size requirements for the measured indicators in this study, we comprehensively assessed and determined that 8 plants per treatment would be used as the sample size for the formal experiment. This number not only meets the basic requirements for sample size in statistical analysis, ensuring the robustness of the results, but also avoid the accumulation of experimental errors caused by an excessive number of samples while taking into account individual differences. we apologize for the lack of clarity regarding experimental replication scheme. We have supplemented the experimental replication scheme in the resubmitted revised manuscript (see revised lines 410-411, 426-427). |
|
Comments 10: The soil moisture level of 55-60% appears too high to be considered a true drought stress treatment in this experimental system. Please justify this threshold or consider adjusting to a lower moisture level that would more realistically induce drought stress. |
|
Response 10: We sincerely apologize for the confusion caused by the statements in the paper. The 55-60% soil moisture level was designated as the well-watered control group (control) in our experimental design, rather than a drought stress treatment. The actual drought stress treatments (D1, D2, D3 and D4) are defined by the soil moisture levels of 30-35%, 25-30%, 20-25% and 15-20%, respectively (see revised lines 401-402). |
|
Comments 11: The drought treatment is inconsistently labeled as D1 (55-60% moisture) in the stress section but referred to as D0 in the ABA mitigation experiment. Please standardize the nomenclature throughout to prevent reader confusion. All sections should use either D0 or D1 consistently for this moisture level. |
|
Response 11: We appreciate your careful attention to the nomenclature of treatments. Upon a thorough review of the manuscript, we confirm that the labeling of stress section is consistent with ABA mitigation experiment. In this study, D0 specifically refers to the control with 55-60% moisture and D1 specifically refers to the drought treatment with 30-35% moisture (see revised lines 401-402, 413). |
|
Comments 12: The experimental design limits causal interpretation of ABA effects, as ABA was only applied to drought-stressed plants. Without parallel ABA treatment in control (well-watered) conditions, observed changes cannot be conclusively attributed to drought-specific ABA responses rather than general ABA effects. |
|
Response 12: We sincerely appreciate your insightful comments. Although limited by the experimental conditions, we were unable to supplement parallel ABA treatments in control (well-watered) conditions. However, we have done our best to mine and analyze from the existing data to address your concerns indirectly. Our experimental design includes treatment groups of well-watered (CK), drought (DR), and drought + ABA (DR+ABA) etc. Although the differences between DR+ABA and CK likely include general ABA effects, but the significant divergence between DR+ABA and DR—particularly in key parameters such as MDA contents and antioxidant enzyme activity—pecifically reflects ABA-mediated mitigation mechanism of drought damage. We sincerely hope the above explanations will meet with your approval. |
|
Thank you again for your valuable time, insightful comments, and meticulous review. Your suggestions have greatly helped improve the rigor and clarity of our manuscript. We have addressed all the points raised in your review and made corresponding revisions as detailed above.
Sincerely, Xiaolan Li |

Reviewer 2 Report
Comments and Suggestions for Authors
Abiotic stress, particularly drought, is a major cause of global crop yield loss. As a result, plants have developed various mechanisms to mitigate such stress. Nitraria tangutorum, a rare wild desert berry, exhibits notable antioxidative and biological activities. This study found that exogenous ABA treatment enhanced the growth of N. tangutorum seedlings, increased leaf RWC, and mitigated photosynthetic inhibition under drought conditions. ABA also boosted osmolyte accumulation and antioxidant enzyme activity, which led to reducing drought-induced damage. The acquired results aim to contribute to the artificial cultivation and ecological restoration of N. Tangutorum and desert ecosystems stability.
Abstract – I d omit the 3rd sentence (line 9) and add more information about the results
Introduction – comprehensive and well written
Results – figures are clearly presenting the results and the statistics is shown.
What is the reason behind choosing this experimental design? You can elaborate in MM section.
Why the cited literature is so old? Arent there any more recent studies you can use? Less than 30% is from the past 5 years.
Author Response
|
Response to Reviewer 2 Comments |
|
Dear Reviewer: Thank you very much for taking the time to review this manuscript entitled “Physiological mechanisms of exogenous ABA in alleviating drought stress in Nitraria tangutorum” (ID: plants-3800017). Those comments are all valuable and very helpful for revising and improving our paper. We have carefully revised the manuscript and provided the point-by-point response below. The changes in the revised manuscript have been highlighted in red. We hope these changes will strengthen our manuscript. |
|
Abstract |
|
Comments 1: Abstract – I d omit the 3rd sentence (line 9) and add more information about the results. |
|
Response 1: We sincerely appreciate your constructive suggestion regarding the abstract structure. Following your recommendation, we have deleted the 3rd sentence (original Line 9) and added more information about the results (see revised lines 13-16). |
|
Introduction |
|
Comments 1: Introduction – comprehensive and well written |
|
Response 1: We appreciate your review of the manuscript and encouraging comment on introduction. |
|
Results |
|
Comments 1: Results – figures are clearly presenting the results and the statistics is shown. |
|
Response 1: We appreciate your review of the manuscript and encouraging comment on results. |
|
Materials and Methods |
|
Comments 1: What is the reason behind choosing this experimental design? You can elaborate in MM section. |
|
Response 1: We sincerely appreciate your valuable suggestion. As recommended, we have added some content in the MM section to clarify the reason for our experimental design (see revised lines 399-401, 414-415). |
|
References |
|
Comments 1: Why the cited literature is so old? Arent there any more recent studies you can use? Less than 30% is from the past 5 years. |
|
Response 1: We sincerely appreciate your valuable suggestions. We have carefully reviewed and reflected the issue of the timeliness of references. The explanation is as follows: N. tangutorum, as a desert-specific shrub, has strong regional and niche characteristics in its research. In recent years, specialized studies on this species have indeed been relatively limited, especially in the aspect of exogenous ABA regulation of its drought response mechanisms. During the literature retrieval process, we have made every effort to incorporate the latest relevant research achievements. Based on your suggestion, we have replaced and updated some of the references in "Materials and Methods". However, as the introduction and discussion involve the analysis of core mechanisms, it is still necessary to rely on classic research to lay a theoretical foundation. We sincerely hope the above explanations will meet with your approval (see the revised references 50-62). |
|
Thank you again for your valuable time, insightful comments, and meticulous review. Your suggestions have greatly helped improve the rigor and clarity of our manuscript. We have addressed all the points raised in your review and made corresponding revisions as detailed above.
Sincerely, Xiaolan Li |

Reviewer 3 Report
Comments and Suggestions for Authors
Overall Comments:
This manuscript reports on physiological responses of N. tangutorum to drought stress. The work also describes some alleviation of negative responses to drought via applications of exogenous ABA. The responses to ABA confirm the typical U-shaped curve of plant hormone responses, so validation of the role of ABA in response to stress is confirmed. None of the physiological parameters measured in response to drought and/or ABA are particularly novel, but documentation of this response in this species appears novel. It is recommended that the authors clarify the responses to exogenous ABA as strictly “survival” oriented, or are there any practical applications of this report to improve production of the species for human use.
Abstract:
The authors claim that this report provides a theoretical basis for rejuvenation and artificial cultivation of N. tangutorum, but there is no discussion on this goal in the paper itself. It is proposed that the authors add some text regarding this concept in the Discussion section. Otherwise, they need to delete this last sentence of the abstract.
Introduction:
Lines 85-87: The same concluding statement from the Abstract is included here at the end of the Introduction, suggesting that the paper will provide evidence of how exogenous ABA could be used in maintenance of desert ecosystems, so the authors must improve the Discussion regarding this point. Otherwise, this statement needs to be removed from the Introduction.
This may require additional data interpretation to project the impact of the physiological changes with ABA on survival and maintenance of biomass accumulation in the species under drought in the desert.
Some shortening of the Intro would be appropriate, deleting several sentences of marginal value. Also, MDA is not mentioned in the Introduction.
Materials and Methods:
Line 379: Please clarify the 1:1 soil mix. What are the two components in this 1:1 mix?
Line 380: What is meant by normal “field management”? I assume this work was done in a greenhouse since the plants are in pots. Please clarify the growing environment description in this paragraph.
Line 389: In this sentence, and throughout the paper, change all times to military time, i.e. 6 pm = 1800 hr.
Line 392: “roots, stems and leaves…. Were mixed and sampled respectively” . Please explain “mixed” or change text.
Line 425-426: Check spelling of malonaldehyde in both lines.
Results:
Figure 1. Very nice Figure.
Figure 2. You do not need the Legend in the upper right corner of each graph since your y-axis title already describes the data set.
Line 117: There was no discussion of MDA in the Introduction, so you need to provide full name of the material prior to use of “MDA”.
Figure 3. Figure legend should contain explanation of R, S and L.
Figure 3a. In this graph and in all others, the word “content” should be replaced with “concentration”. Content infers “amount in an organ or tissue fraction” while concentration better describes “amount per unit mass”.
Figure 5. Best data set in the paper. Authors should expand their discussion of this data to propose any practical application of exogenous ABA on growth, such as establishment of seedlings, or spraying ABA on established plants prior to an extreme drought event to improve survival. Also, this figure confirms a U-shaped hormonal response, which confirms biological theory.
Line 205-207: This statement is a stretch, since Fo and Fv/Fm were not positively impacted at all by the exogenous ABA treatments vs. CKD. Please clarify in more detail.
Line 252-255: The statement implies more positive response than the data demonstrate. Only DA2 changed starch in L only, not other tissues.
Discussion:
Conclusions:
It is proposed that the authors provide some practical application of their data to improving production of the species under in situ desert conditions. That was the main stated objective of the work, and there needs to be some conclusion regarding interpretation of the data towards that end.
Author Response
|
Response to Reviewer 3 Comments |
|
Dear Reviewer: Thank you very much for taking the time to review this manuscript entitled “Physiological mechanisms of exogenous ABA in alleviating drought stress in Nitraria tangutorum” (ID: plants-3800017). Those comments are all valuable and very helpful for revising and improving our paper, as well as the important guiding significance to our researches. We have carefully revised the manuscript and provided the point-by-point response below. The changes in the revised manuscript have been highlighted in red. We hope these changes will strengthen our manuscript. |
|
Abstract: |
|
Comments 1: The authors claim that this report provides a theoretical basis for rejuvenation and artificial cultivation of N. tangutorum, but there is no discussion on this goal in the paper itself. It is proposed that the authors add some text regarding this concept in the Discussion section. Otherwise, they need to delete this last sentence of the abstract. |
|
Response 1: We sincerely appreciate the valuable comments. We agree with this comment. Therefore, we have modified the sentence “This study provides a theoretical basis for rejuvenation and artificial cultivation of N. tangutorum” in the abstract to “This study provides a theoretical basis for restoration, propagation and protection of N. tangutorum” and added relevant content in the discussion (see revised lines 17-18, 380-383). |
|
Introduction: |
|
Comments 1: Lines 85-87: The same concluding statement from the Abstract is included here at the end of the Introduction, suggesting that the paper will provide evidence of how exogenous ABA could be used in maintenance of desert ecosystems, so the authors must improve the Discussion regarding this point. Otherwise, this statement needs to be removed from the Introduction. This may require additional data interpretation to project the impact of the physiological changes with ABA on survival and maintenance of biomass accumulation in the species under drought in the desert. |
|
Response 1: We sincerely apologize for the oversight in our original manuscript. We have carefully revised the relevant statements in both the Abstract and Introduction to ensure they are more in line with the research findings. We have also supplemented and expanded the content of the discussion section to provide a more in-depth and well-supported argument for this point of view (see revised lines 85-86, 380-383). |
|
Comments 2: Some shortening of the Intro would be appropriate, deleting several sentences of marginal value. Also, MDA is not mentioned in the Introduction. |
|
Response 2: We sincerely appreciate the reviewer’s insightful observation regarding the layout of the Introduction. We have carefully shortened the Introduction by removing several sentences of marginal relevance. Additionally, we have incorporated a discussion on MDA in the Introduction (see revised lines 54-57). |
|
Materials and Methods: |
|
Comments 1: Line 379: Please clarify the 1:1 soil mix. What are the two components in this 1:1 mix? |
|
Response 1: We sincerely thank you for careful reading. We apologize for any lack of clarity in our original description. Our intended meaning was to mix completely air-dried sand and soil in a volume ratio of 1:1 as the culture medium. We have revised the relevant content in our resubmitted manuscript (see revised lines 392-394). |
|
Comments 2: Line 380: What is meant by normal “field management”? I assume this work was done in a greenhouse since the plants are in pots. Please clarify the growing environment description in this paragraph. |
|
Response 2: We sincerely appreciate the reviewer's careful reading and valuable comments. We apologize for the lack of accuracy in our original description. This work was indeed conducted in plastic greenhouses at Gansu Agricultural University, not under open field conditions. The term 'normal field management' includes daily watering and pest control measures as required. We have revised the relevant content in the resubmitted manuscript to explicitly state that the plants were cultivated using standard pot-culture management (see revised lines 394-395). |
|
Comments 3: Line 389: In this sentence, and throughout the paper, change all times to military time, i.e. 6 pm = 1800 hr. |
|
Response 3: We sincerely appreciate the reviewer's suggestion regarding time notation standardization. We have carefully revised all times throughout the manuscript to comply with military time format as recommended (see revised lines 405, 421, 422). |
|
Comments 4: Line 392: “roots, stems and leaves…. Were mixed and sampled respectively” . Please explain “mixed” or change text. |
|
Response 4: We appreciate your correction of the term "mixed". We have changed "mixed" to "pooled" in the revised manuscript (see revised lines 408, 425). |
|
Comments 5: Line 425-426: Check spelling of malonaldehyde in both lines. |
|
Response 5: We sincerely appreciate you for identifying this spelling inconsistency. We apologize for our oversight. We have carefully reviewed and uniformly corrected this term to the standard spelling throughout the manuscript (see revised lines 443). |
|
Results: |
|
Comments 1: Figure 1. Very nice Figure. |
|
Response 1: We appreciate your review of the manuscript and encouraging comment on Figure 1. |
|
Comments 2: Figure 2. You do not need the Legend in the upper right corner of each graph since your y-axis title already describes the data set. |
|
Response 2: We sincerely appreciate your suggestion regarding Figure 2. As recommended, we have removed the redundant legends from Figures 2 and 6 (see revised figures 2 and 6). |
|
Comments 3: Line 117: There was no discussion of MDA in the Introduction, so you need to provide full name of the material prior to use of “MDA”. |
|
Response 3: We appreciate your valuable comments. We have add a discussion of MDA in the Introduction and carefully reviewed the full text to ensure that each abbreviation is clearly defined when it is first mentioned (see revised lines 54-57). |
|
Comments 4: Figure 3. Figure legend should contain explanation of R, S and L. |
|
Response 4: We sincerely appreciate this constructive suggestion. We have explained in the relevant figure legends that R, S, and L respectively represent roots, stems, and leaves (see revised lines 148、181、259 and 290). |
|
Comments 5: Figure 3a. In this graph and in all others, the word “content” should be replaced with “concentration”. Content infers “amount in an organ or tissue fraction” while concentration better describes “amount per unit mass”. |
|
Response 5: We sincerely appreciate your professional guidance. We have systematically replaced "content" with "concentration" throughout all figures (see revised figures). |
|
Comments 6: Figure 5. Best data set in the paper. Authors should expand their discussion of this data to propose any practical application of exogenous ABA on growth, such as establishment of seedlings, or spraying ABA on established plants prior to an extreme drought event to improve survival. Also, this figure confirms a U-shaped hormonal response, which confirms biological theory. |
|
Response 6: We sincerely appreciate your valuable suggestion regarding Figure 5. Following your advice, we have expanded the discussion on this data set in the revised manuscript (see revised 344-349). |
|
Comments 7: Line 205-207: This statement is a stretch, since Fo and Fv/Fm were not positively impacted at all by the exogenous ABA treatments vs. CKD. Please clarify in more detail. |
|
Response 7: We sincerely appreciate your careful review and pointed-out issue. Upon rechecking the data, we regret to find that there was an error in marking the significance letters for Fo and Fv/Fm in the Figure 6, which led to the misleading statement. After correcting the significance markers, the data actually show that exogenous ABA treatments did have a positive impact on Fo and Fv/Fm compared to the CKD group. We have revised the Figure 6 in the revised manuscript to accurately reflect this trend, and we apologize for the confusion caused by our oversight (see revised Figure 6). |
|
Comments 8: Line 252-255: The statement implies more positive response than the data demonstrate. Only DA2 changed starch in L only, not other tissues. |
|
Response 8: We sincerely appreciate your valuable comments. We have revised the descriptions of Figure 8A to strictly align with the experimental results (see revised 248-254). |
|
Conclusions: |
|
Comments 1: It is proposed that the authors provide some practical application of their data to improving production of the species under in situ desert conditions. That was the main stated objective of the work, and there needs to be some conclusion regarding interpretation of the data towards that end. |
|
Response 1: We sincerely appreciate your guidance on connecting our findings to practical applications under in situ desert conditions. Following your suggestion, we have expanded the Conclusion section to explicitly address the practical implications of our data for improving the production of N. tangutorum in desert environments (see revised lines 505-510). |
|
Thank you again for your valuable time, insightful comments, and meticulous review. Your suggestions have greatly helped improve the rigor and clarity of our manuscript. We have addressed all the points raised in your review and made corresponding revisions as detailed above.
Sincerely, Xiaolan Li |

Round 2
Reviewer 1 Report
Comments and Suggestions for Authors
The authors have made relevant revisions to the manuscript, resulting in a certain degree of improvement in its quality. They have also provided relatively thorough responses and explanations regarding some deficiencies in the experimental design.
Author Response
Dear Reviewer,
We would like to express our sincere gratitude for your recognition and affirmation of the revisions made to our manuscript. Your positive feedback serves as significant encouragement for our team to further refine our research work.
We will continue to uphold a rigorous attitude and carefully address any potential detailed optimizations that may be involved in the subsequent stages of the manuscript. We also look forward to advancing the improvement of our research results with your professional guidance.
Once again, thank you for the precious time and professional review you have dedicated!
Sincerely,
Xiaolan Li